# Adenosine A_2A_ Receptor Antagonists Affects NMDA Glutamate Receptor Function. Potential to Address Neurodegeneration in Alzheimer’s Disease

**DOI:** 10.3390/cells9051075

**Published:** 2020-04-26

**Authors:** Rafael Franco, Rafael Rivas-Santisteban, Mireia Casanovas, Alejandro Lillo, Carlos A. Saura, Gemma Navarro

**Affiliations:** 1Departament de Bioquímica i Biomedicina Molecular, Universitat de Barcelona, 08028 Barcelona, Spain; rrivasbioq@gmail.com (R.R.-S.); mcasanfe8@gmail.com (M.C.); 2Centro de Investigación Biomédica en Red sobre Enfermedades Neurodegenerativas, Instituto de Salud Carlos III, Valderrebollo, 5, 28031 Madrid, Spain; carlos.saura@uab.cat; 3Departament of Biochemistry and Physiology, Faculty of Pharmacy and Food Science, Universitat de Barcelona, 08028 Barcelona, Spain; alilloma55@gmail.com; 4Institut de Neurociències, Department de Bioquímica i Biologia Molecular, Universitat Autònoma de Barcelona, Bellaterra, 08193 Barcelona, Spain

**Keywords:** G-protein-coupled receptors, functional selectivity, microglia, neuroprotection, cognition, signaling

## Abstract

(1) Background. *N*-methyl d-aspartate (NMDA) ionotropic glutamate receptor (NMDAR), which is one of the main targets to combat Alzheimer’s disease (AD), is expressed in both neurons and glial cells. The aim of this paper was to assess whether the adenosine A_2A_ receptor (A_2A_R), which is a target in neurodegeneration, may affect NMDAR functionality. (2) Methods. Immuno-histo/cytochemical, biophysical, biochemical and signaling assays were performed in a heterologous cell expression system and in primary cultures of neurons and microglia (resting and activated) from control and the APP_Sw,Ind_ transgenic mice. (3) Results. On the one hand, NMDA and A_2A_ receptors were able to physically interact forming complexes, mainly in microglia. Furthermore, the amount of complexes was markedly enhanced in activated microglia. On the other hand, the interaction resulted in a novel functional entity that displayed a cross-antagonism, that could be useful to prevent the exacerbation of NMDAR function by using A_2A_R antagonists. Interestingly, the amount of complexes was markedly higher in the hippocampal cells from the APP_Sw,Ind_ than from the control mice. In neurons, the number of complexes was lesser, probably due to NMDAR not interacting with the A_2A_R. However, the activation of the A_2A_R receptors resulted in higher NMDAR functionality in neurons, probably by indirect mechanisms. (4) Conclusions. A_2A_R antagonists such as istradefylline, which is already approved for Parkinson’s disease (Nouriast^®^ in Japan and Nourianz^®^ in the US), have potential to afford neuroprotection in AD in a synergistic-like fashion. i.e., via both neurons and microglia.

## 1. Introduction

Alzheimer’s disease (AD) is characterized by two pathological hallmarks, amyloid plaques composed of β-amyloid peptides and neurofibrillary tangles composed of hyperphosphorylated tau protein. Patients are currently treated with two drug types: *N*-methyl d-aspartate ionotropic glutamate receptor (NMDAR) modulators and acetylcholinesterase inhibitors. An anti-AD approved drug acting on NMDAR, memantine (marketed in many Countries as Namenda^®^), is a negative allosteric modulator acting as a low-affinity open channel blocker. The drug was developed to find a weak negative modulator as both strong or weak NMDAR activities are noxious [1,2,3]. Unfortunately, the efficacy of current drugs is low and none prevent the disease progression [4]. It is well established that AD correlates with NMDAR functional alterations that cannot be addressed by drugs acting in the orthosteric center; in fact, NMDAR-related drugs used in AD patients are allosteric modulators of the receptor.

Adenosine is an autacoid, i.e., a hormone-like locally acting molecule, that exerts metabolic and regulatory functions in almost any tissue and cell type of the mammalian body. In the brain, it is one of the main neuromodulators acting via four G-protein-coupled receptors (GPCRs): A_1_, A_2A_, A_2B_ and A_3_. The adenosine A_2A_ receptor (A_2A_R), a G_s_-coupled GPCR, is heavily expressed in the motor control brain areas [5] but is also expressed in other CNS regions [5]. Remarkably, an antagonist of the receptor, istradefylline, has been recently approved as a first-in-class drug for the treatment of Parkinson’s disease [6,7,8] (Nouriast^®^ in Japan and Nourianz^®^ in the US). Apart from combating motor symptoms and/or minimizing the side effect of anti-parkinsonian medication, A_2A_R antagonists may be neuroprotective. Mechanisms of neuroprotection are based in in vitro and in vivo pharmacological studies. Pioneering work using adenosine itself led to a review entitled “Cerebral Protection by Adenosine” [9]. Based on data using brain ischemia-reperfusion models, it was already evident that adenosine could achieve the same neuroprotective effects as NMDAR blockers. The data came from studies focused on neurons, and the molecular mechanisms, including the type of adenosine receptor, were not known. As pointed out below, NMDAR are expressed in glia where adenosine receptors are also expressed. It is noteworthy that the A_2A_R is upregulated in activated microglia [10].

More recent studies conducted to explore the therapeutic possibilities to combat AD have used different assays with genetic ablation or a pharmacological blockade of the adenosine A_2A_ receptor, resulting in neuroprotection in different models. Briefly, taking into account the most recent reports, the expression of the A_2A_ receptor is altered even in peripheral blood cells from patients with AD or vascular dementia [11]. In addition, deletion of the receptor was beneficial in a tauopathy mouse model [12] and A_2A_R antagonists were protective in both the APPswe/PS1dE9 [13] and the triple 3×Tg-AD transgenic AD models [14]. Finally it should be noted that the early synaptic events seemed mediated by the A_2A_R in the 3×Tg-AD transgenic AD mouse model [15].

Based on the extensive background, the aim of this paper was to investigate whether the activation of the A_2A_R may regulate NMDAR function in both neurons and microglia cells. Although NMDAR function is instrumental for neurotransmission, senescent neurons have little resources to prevent death and rely on the support provided by glial cells. Among them, microglia are of interest since A_2A_R expression is enhanced in activated microglia where the NMDAR is also expressed [16,17,18,19]. Our results suggested that A_2A_R antagonists may provide neuroprotection via the modulation of NMDAR functionality.

## 2. Materials and Methods

### 2.1. Reagents

Lipopolysaccharide (LPS) and interferon-γ (IFN-γ) were purchased from Sigma Aldrich (St Louis, MO, USA); receptor ligands were purchased from Tocris Bioscience (Bristol, UK). Agonists were: *N*-methyl d-aspartate (NMDA) 4-[2-[[6-Amino-9-(*N*-ethyl-β-d-ribofuranuronamidosyl)-9*H*-purin-2-yl]amino]ethyl]benzenepropanoic acid hydrochloride (CGS-21680). Antagonists were: (*5S*,*10R*)-(+)-5-Methyl-10,11-dihydro-5*H*-dibenzo[a,d]cyclohepten-5,10-imine maleate (MK-801) and 2-(2-Furanyl)-7-(2-phenylethyl)-7*H*-pyrazolo[4,3-e][1,2,4]triazolo[1,5-c]pyrimidin-5-amine (SCH-58261).

### 2.2. Fusion Proteins

To create A_2A_R-*Renilla* luciferase (RLuc and A_2A_R-yellow fluorescent protein (YFP) fusion molecules, the human version of adenosine A_2A_R cDNA lacking the stop codon, was obtained by PCR and subcloned to a *Renilla* luciferase (RLuc)-containing vector (p*RLuc*; PerkinElmer, Wellesley, MA, USA) and a yellow fluorescent protein (YFP)-containing vector (pEYEP-N1; Clontech, Heidelberg, Germany) using sense and antisense primers harboring unique restriction sites for HindIII and BamHI. A similar approach was used to generate the cDNAs for GluN1-RLuc and caldendrin-YFP fusion proteins.

### 2.3. APP Transgenic Mouse Model of Alzheimer’s Disease (AD)

APP_Sw,Ind_ transgenic mice (line J9; C57BL/6 background), expressing human APP695 harboring the familial AD-linked Swedish (K670N/M671L) and Indiana (V717F) mutations under the PDGFβ promoter, were obtained by crossing APP_Sw,Ind_ to non-transgenic (WT) mice [20]. Animals come from a colony established by co-author Carlos A. Saura in the Autonomous Barcelona University; animals came directly from the laboratory who developed the transgenic animal after signing the *ad hoc* Transfer Agreement.

### 2.4. Cell Culture and Transfection

HEK-293T human embryonic kidney cells from the American Type Culture Collection (ATCC) were grown in Dulbecco’s modified Eagle’s medium (DMEM) (Gibco, Paisley, Scotland, UK) supplemented with 2 mM l-glutamine, 100 U/mL penicillin/streptomycin, MEM Non-Essential Amino Acids Solution (1/100) and 5% (*v*/*v*) heat inactivated fetal bovine serum (FBS) (Gibco, Paisley, Scotland, UK). The cells were maintained in a humid atmosphere of 5% CO_2_ at 37 °C. The cells were transiently transfected using Polyethylenimine (PEI, Sigma Aldrich, St. Louis, MO, USA) as previously described [21].

To prepare mice primary microglial cultures (C57BL/6 wild type or transgenic mice), the brain was removed at postnatal days 2 to 4. The microglial cells were isolated as described in [22] and grown in a DMEM medium supplemented with 2 mM l-glutamine, 100 U/mL penicillin/streptomycin, and 5% (*v*/*v*) heat inactivated FBS. For neuronal primary cultures, the hippocampus from mouse embryos (E19) was removed and the neurons were isolated as described by Hradsky et al., 2013 [23]. Cells were grown in a neurobasal medium supplemented with 2 mM l-glutamine, 100 U/mL penicillin/streptomycin, supplement (2% *v*/*v*) with B27 (Gibco). For cAMP assays, cells were grown on 6-well plates at a density of 500,000 cells/well, for ERK 1/2 phosphorylation assays, cells were placed in 96-well plates at a density of 50,000 cells/well; for proximity ligation assay cells were placed in 12-well plates with coverslips. Cell counting was assessed using trypan blue and a countless II FL automated cell counter (Thermo Fisher Scientific, Waltham, MA, USA). Experiments were carried out 15 days later and the medium was replaced every 5–7 days. The animal handling and protocols were conducted in accordance with the European Council Directive 2010/63/UE as well as in keeping with the current Spanish legislation (RD53/2013). The ethics committee of the two institutions (University of Barcelona and Autonomous University of Barcelona) were in charge of law implementation.

### 2.5. Immunocytochemistry

The transfected HEK-293T cells or primary microglial culture cells seeded in coverslips were fixed in 4% paraformaldehyde for 15 min and washed twice with phosphate-buffered saline (PBS) containing 20 mM glycine, before permeabilization with PBS-glycine containing 0.2% Triton X-100 (5 min incubation for the HEK-293T cells and 15 min for the microglial culture cells). The HEK-293T cells were treated for 1 h with PBS containing 1% bovine serum albumin (BSA), labeled with mouse monoclonal anti-RLuc antibody (1/100; mAB4400, EMD Millipore, Darmstadt, Germany) and subsequently treated with Cy3 anti-mouse (1/200; 715-166-150, Jackson ImmunoResearch (red)) immunoglobulin G (IgG) (1 h each). Microglial cells were treated for 1 h with PBS containing 1% BSA and labelled with a mouse anti-iNOS (1/100; NOS2 (C-11): sc-7271; SCB) antibody, a mouse monoclonal anti-arginase I (1/100; 610708; BD Biosciences, San Jose, CA, USA) antibody or a rabbit polyclonal anti-Ki-67 (1/100; ab15580; Abcam, Cambridge, UK) antibody, and subsequently treated with a Cy3-conjugated anti-rabbit (1/200; 711-165-152; Jackson ImmunoResearch (red), West Grove, PA, USA) or anti-mouse (1/200; 715-166-150; Jackson ImmunoResearch (red), West Grove, PA, USA) IgG secondary antibodies (1 h each). The nuclei were stained with Hoechst (1/100; Sigma Aldrich, St. Louis, MO, USA). The samples were washed several times and mounted with 30% Mowiol (Calbiochem, San Diego, CA, USA). The images were obtained in a Leica SP2 confocal microscope (Leica Microsystems). The instrument was equipped with an apochromatic 63X oil-immersion objective (N.A. 1.4), and 488 nm and 561 nm laser lines.

### 2.6. Bioluminescence Resonance Energy Transfer (BRET) Assays

For the BRET assays, the HEK-293T cells were transiently co-transfected with a constant amount of cDNAs encoding for GluN1-RLuc and GluN2 and with increasing amounts of cDNAs corresponding to A_2A_R-YFP or caldendrin-YFP. Forty-eight hours post-transfection, the cell suspension was adjusted to 20 μg of protein using a Bradford assay kit (Bio-Rad, Munich, Germany) and BSA for standardization. To quantify the protein-YFP expression, fluorescence was read in a Mithras LB 940 (Berthold Technologies, Bad Wildbad, Germany) equipped with a high-energy xenon flash lamp, using a 10 nm bandwidth excitation filter at 485 nm reading. For BRET and BRET with bimolecular complementation (BiFLC) measurements, the readings were collected 1 min after the addition of 5 μM coelenterazine H (Molecular Probes, Eugene, OR, USA) using a Mithras LB 940, which allowed the integration of the signals detected in the short-wavelength filter at 485 nm and the long-wavelength filter at 530 nm. To quantify the protein-RLuc expression, luminescence readings were performed 10 min after the 5 μM coelenterazine H addition using a Mithras LB 940. The net BRET was defined as ((long-wavelength emission)/(short-wavelength emission)) − C_f_, where C_f_ corresponds to ((long-wavelength emission)/(short-wavelength emission)) for the donor construct expressed alone in the same experiment. The GraphPad Prism software (San Diego, CA, USA) was used to fit the data. BRET is expressed as milli BRET units, mBU (net BRET × 1000).

### 2.7. Cyclic Adenylic Acid (cAMP) Determination

Two hours before initiating the experiment, culture medium for HEK-293T-transfected or primary neuronal or glial cells was exchanged by serum-starved DMEM medium. Then, the cells were detached, resuspended in a growing medium containing 50 µM zardaverine (Tocris Bioscience, Bristol, UK) and plated in 384-well microplates (2500 cells/well), pretreated (15 min) with the corresponding antagonists (SCH-58261 for A_2A_R and MK-801 for NMDAR) or vehicle and stimulated with agonists (CGS-21680 for A_2A_R and NMDA for NMDAR) (15 min) before adding 0.5 μM forskolin or vehicle (15 min). The readings were performed after a 1 h incubation (room temperature). Homogeneous time-resolved fluorescence energy transfer (HTRF) measures were performed using the Lance Ultra cAMP kit (PerkinElmer, Waltham, MA, USA). Fluorescence at 665 nm was analyzed on a PHERAstar Flagship microplate reader equipped with an HTRF optical module (BMG Lab technologies, Offenburg, Germany).

### 2.8. Extracellular Signal-Regulated Kinase (ERK) Phosphorylation Determination

To determine the ERK1/2 phosphorylation, 40,000 HEK-293T cells/well, 50,000 microglia cells/well or 50,000 neurons/well were plated in transparent 96-well microplates and kept in the incubator for 48 h (HEK-293T cells) or 12 days (microglia and neuronal culture cells). Two to four h before initiating the experiment, the medium was substituted for a serum-starved DMEM medium. Then, the cells were pre-treated at room temperature for 10 min with the specific antagonists (SCH-58261 for A_2A_R and MK-801 for NMDAR) or vehicle in a serum-starved DMEM medium and stimulated for an additional 10 min with the specific agonists (CGS-21680 for A_2A_R and *N*-methyl d-aspartate (NMDA) for NMDAR) or vehicle. The cells were then washed twice with cold PBS before the addition of a lysis buffer (20 min treatment in constant agitation). Subsequently, 10 μL of each supernatant was placed in white ProxiPlate 384-well microplates and the ERK 1/2 phosphorylation was determined using the AlphaScreen^®^SureFire^®^ kit (Perkin Elmer, Waltham, MA, USA) following the instructions of the supplier and using an EnSpire^®^ Multimode Plate Reader (PerkinElmer, Waltham, MA, USA).

### 2.9. Assessment of Dynamic Mass Redistribution (DMR)

The cell mass redistribution induced upon receptor activation was detected by illuminating the underside of a biosensor with a polychromatic light and measuring the changes in the wavelength of the reflected monochromatic light that was a sensitive function of the index of refraction. The magnitude of this wavelength shift (in picometers) was directly proportional to the amount of DMR. HEK-293T cells and neuronal and microglial primary cultures were seeded in 384-well sensor microplates to obtain 70–80% confluent monolayers constituted of approximately 10,000 cells per well. Prior to the assay, the cells were washed twice and incubated for 2 h with assay buffer (Hank’s balanced salt solution (HBSS) with 20 mM 4-(2-hydroxyethyl)-1-piperazineethanesulfonic acid (HEPES) buffer, pH 7.15) containing 0.1% DMSO (24 °C, 30 μL/well). Hereafter, the sensor plate was scanned and a baseline optical signature was recorded for 10 min before adding the 10 μL of the specific antagonists (SCH-58261 for A_2A_R and MK-801 for NMDAR), that were recoded for 30 min followed by the addition of 10 μL of the specific agonists (CGS-21680 for A_2A_R and NMDA for NMDAR); all the test compounds were dissolved in the assay buffer. The cell signaling signature was determined using an EnSpire^®^ Multimode Plate Reader (PerkinElmer, Waltham, MA, USA) by a label-free technology. The results were analyzed using the EnSpire Workstation Software v 4.10.

### 2.10. Determination of Cytoplasmic Calcium Ion Level Increase

HEK-293T cells were transfected with the cDNAs for human A_2A_R, for GCaMP6 calcium sensor, and/or for both the GluN1 and GluN2 subunits of the NMDAR [24]. Forty-eight hours after transfection, 150,000 HEK-293T cells/well were plated in 96-well black, clear bottom microtiter plates and were incubated with Mg^2+^-free Locke’s buffer (154 mM NaCl, 5.6 mM KCl, 3.6 mM NaHCO_3_, 2.3 mM CaCl_2_, 5.6 mM glucose and 5 mM HEPES, pH 7.4) supplemented with 10 μM glycine. The cells were treated with the specific antagonists (SCH-58261 for A_2A_R and MK-801 for NMDAR) for 10 min, followed by the addition of the receptor agonists, CGS-21680 for A_2A_R and NMDA for NMDAR, just a few seconds before the readings. The fluorescence emission intensity of the GCaMP6 was recorded at 515 nm upon excitation at 488 nm on the EnSpire^®^ Multimode Plate Reader (PerkinElmer, Waltham, MA, USA) for 225 s every 5 s.

### 2.11. In Situ Proximity Ligation Assay (PLA)

PLA was performed with reagents from Sigma Aldrich and following the protocols of the supplier. In brief, microglial and neuronal primary cells grown on glass coverslips were fixed in 4% paraformaldehyde for 15 min, washed with PBS containing 20 mM glycine to quench the aldehyde groups, and permeabilized with the same buffer containing 0.05% Triton X-100 (15 min). After 1 h incubation at 37 °C with blocking solution, the cells were treated with specific antibodies against A_2A_ or NMDA receptors: mouse monoclonal anti-A_2A_R (1/100, Millipore, Darmstadt, Germany) or rabbit polyclonal anti-GluN1 antibody (1/200, Millipore). The cells were processed using the PLA probes detecting mouse and rabbit antibodies (Duolink II PLA probe anti-rabbit plus and Duolink PLA probe anti-mouse minus; Sigma Aldrich) and were prepared with Hoechst (1/200; Sigma Aldrich, St. Louis, MO, USA) using a mounting medium. The images were obtained in a Leica SP2 confocal microscope (Leica Microsystems, Mannheim, Germany). For each field of view a stack of two channels (one per staining) and 3 to 4 Z stacks with a step size of 1 µm were acquired. The quantification of the cells containing one or more red spots versus the total cells (blue nucleus), and in cells containing spots, the ratio r (number of red spots/cell), were determined by the Duolink Image tool software (Sigma Aldrich, St. Louis, MO, USA).

### 2.12. Statistical Analysis

The data in the graphs are the mean ± SEM (*n* = 5, at least). The GraphPad Prism software version 7 (San Diego, CA, USA) was used for the data fitting and statistical analysis. The Kolmogorov-Smirnov test with the correction of Lilliefors was used to evaluate the normal distribution and the Levene test was used to evaluate the homogeneity of variance. A one-way ANOVA followed by the post-hoc Bonferroni’s test were used when comparing multiple values. When a pair of values were compared, the Student’s t test was used. Significant differences were considered when the *p* value was < 0.05.

## 3. Results

### 3.1. NMDA Receptors May Directly Interact with Adenosine A_2A_ Receptors

*N*-methyl-d-aspartate receptor (NMDAR) activation regulates synaptic plasticity and neuronal survival. However, an increase in the NMDAR activity is associated to excitotoxicity and cell death and this is the main reason it is targeted by one of the existing anti-AD drugs. On the other hand, adenosine is a neuromodulator and one of the main mediators is the A_2A_ receptor (A_2A_R), which is expressed in both CNS neurons and glial cells. To address a potential interaction between the two receptors, we first performed immunocytochemical assays in a heterologous expression system. HEK-293T cells were transfected with cDNAs for A_2A_R-RLuc or for GluN1 fused to RLuc and the GluN2B subunit of NMDAR (transfection of the two subunits was necessary for reconstituting a functional NMDA receptor). The membrane and cytoplasmic expression of both: A_2A_ and NMDA receptors was observed (Figure 1A). Then, colocalization was addressed in cells transfected with the cDNAs for A_2A_R-YFP and for GluN1-RLuc and GluN2B subunits. The degree of colocalization was significant (Figure 1B) but while it demonstrated expression in the same compartment(s), it did not allow the discovery of direct protein–protein interaction. Accordingly, the A_2A_-NMDA receptor interaction was assayed by bioluminescence resonance energy transfer (BRET) using HEK-293T cells expressing a reconstituted NMDAR fused to RLuc and increasing amounts of A_2A_R-YFP. The saturable BRET curve indicated a specific interaction between the A_2A_ and the NMDA receptors (BRET_max_ 141 ± 14 and BRET_50_ 155 ± 22). As a negative control, the A_2A_R was substituted by the calcium sensor protein, caldendrin, and the result was a linear relationship that was indicative of unspecificity (Figure 1C). In summary A_2A_R may interact with NMDAR but not with caldendrin in living transiently transfected HEK-293T cells.

### 3.2. Functional Properties of A_2A_-NMDA Receptor Heteromer Complex

Adenosine A_2A_R couple to G_s_ proteins, activating adenylate cyclase and increasing cAMP intracellular levels. In preliminary assays, we demonstrated that cytosolic cAMP levels increased when HEK-293T cells expressing A_2A_R were treated with the selective ligand, CGS-21680 (Appendix A). Furthermore, this effect was specific, because it was blocked by pre-treatment with the selective antagonist SCH-58261. Neither NMDA, nor a NMDAR antagonist, MK-801, induced any effect (Appendix A). Similar results were obtained in extracellular signal-regulated kinase (ERK) phosphorylation and label-free dynamic mass redistribution (DMR) assays (Appendix A). As expected, cytosolic calcium did not increase upon A_2A_R stimulation (Appendix A) but it did increase when HEK-293T cells were transfected with reconstituted NMDAR and treated with NMDA (Appendix A). Whereas the stimulation with NMDA did not alter the cytosolic cAMP levels (Appendix A), it induced MAPK phosphorylation and modified the DMR outputs. Although the effects were specific and blocked by a selective NMDAR antagonist, MK-801 (Appendix A), they were not altered by either a selective A_2A_R agonist, CGS-21680, or a selective A_2A_R antagonist, SCH-58261.

Once receptor-mediated signaling was characterized in the cells expressing A_2A_R or NMDAR receptors, cross-modulation was assayed in the co-transfected HEK-293T cells in which the cAMP levels, MAPK phosphorylation, label-free DMR and the calcium release signals were analyzed. Remarkably, the cAMP data revealed that the signal obtained after the A_2A_R stimulation was blocked by both A_2A_R and NMDAR antagonists (Figure 2A). Such a property is known as cross-antagonism and would be useful to identify A_2A_-NMDA receptor complexes in natural sources. In fact, cross-antagonism is considered a heteromer print [25,26]. Coactivation of the A_2A_ and NMDA receptors led to a decrease in the CGS-21680-induced effect, indicating that the NMDAR activation impacted on adenosine A_2A_R signaling. In the MAPK phosphorylation assays, the activation of both A_2A_ and NMDA receptors were able to activate the MAPK pathway. In addition, whereas the A_2A_R-mediated signaling was smaller when the two receptors were co-expressed, the NMDAR-mediated signaling was stronger. This result indicated a possible potentiation of A_2A_R over the NMDAR signaling when forming the A_2A_-NMDA receptor complexes. However, the coactivation of both receptors did not result in an additive effect. It is noteworthy that any signal was counteracted by either the NMDAR or A_2A_R selective antagonists, i.e., cross-antagonism was also detected. (Figure 2B). DMR data were similar to those found in cAMP and MAPK assays. The A_2A_R- and NMDAR-agonist-induced signals were blocked by both the NMDAR and A_2A_R selective antagonists, while the coactivation of both receptors produced a stronger signal than that of the NMDA but smaller than that of the A_2A_R agonist. Finally, in terms of calcium mobilization, the coactivation of both receptors produced a similar effect to that induced by the NMDAR stimulation but interestingly, the NMDA effect was blocked by both NMDA and A_2A_ receptor antagonists (A_2A_ receptor agonists did not yet induce cytosolic calcium increases) (Figure 2D). The results may be explained by i) the A_2A_R expression increasing the NMDAR function and ii) the NMDAR activation blocking adenosine A_2A_R signaling and iii) a cross-antagonism due to inter-protomer allosteric communication within the A_2A_R and NMDAR heteromer.

### 3.3. A_2A_-NMDA Receptor Heteromer Complex Expression in Resting and Activated Microglia

Due to the renowned interest in glial cells as targets to combat neurodegenerative diseases and due to the expression of both receptors in these cells, we next analyzed the A_2A_-NMDA receptor complex expression in the microglial primary cultures from wild type mice. Interestingly, the proximity ligation assay (PLA) showed 23% of cells with clusters of receptor complexes depicted as red dots (two red dots per cell) (Figure 3A,B). Moreover, when the microglia was activated using LPS and IFN-γ, the A_2A_R-NMDAR heteromer expression was markedly enhanced, with 92% of cells showing red dots and an eight-fold increase in dots/cell (16 dots in activated cells versus two in resting cells) (Figure 3A,B). Similar assays were performed in hippocampal neuronal primary cultures; 28% of neurons showed red dots. Overall, these results showed that the A_2A_-NMDA receptor complexes may play a relevant role in activated microglia cells.

The functional cross-talk was assayed by the analysis of the cAMP intracellular levels in the microglia cells and in neurons. In both, the resting and the activated microglia, only the activation of the A_2A_R results in cAMP responses, that were reverted by the A_2A_R antagonist and by the NMDAR antagonist. We observed similar results in neurons but without cross-antagonism, probably reflecting that not all NMDA receptors were interacting with the A_2A_R (Figure 3C–E). The activation of either the NMDAR or the A_2A_R resulted in the MAPK pathway activation. Interestingly, coactivation led to a small non additive effect only in microglia cultures, once more supporting the idea that not all NMDA receptors in neurons were interacting with the A_2A_R. However, cross-antagonism was detected in both the microglial and neuronal cells (Figure 3F–H).

### 3.4. A_2A_R-NMDAR Heteromer Expression is Elevated in Primary Microglia from APP_Sw/Ind_ Mice

Alzheimer’s disease (AD) is one of the most prevalent neurodegenerative diseases worldwide; the causes are unknown and current drugs have little efficacy. NMDAR function is enhanced in the initial stages of AD, thus leading to altered intracellular calcium handling and the gradual loss of synaptic function [27,28]. In addition, A_2A_R expression markedly increases in neurodegenerative diseases with an inflammatory component. Accordingly, we moved to assess the expression of A_2A_R-NMDAR heteromers in a transgenic AD mouse model.

We first analyzed the expression of A_2A_-NMDA receptor complexes in the primary cultures of microglia and neurons from the hippocampus of APP_Sw,Ind_ mice. The hippocampal microglia and neuronal primary cultures were obtained from two-day-old pups or 19-day-old fetuses, respectively, and were independently cultured. PLA data analysis was blindly done before knowing the genotyping results. Remarkably, whereas 29% of the cultured microglia from the controls animals showed red dots, the percentage of cells from transgenic animals was 75% (three dots/cell in control versus six dots/cell in transgenic) (Figure 4A,B). The same assay type in neurons led to a low percentage of cells expressing receptor clusters (11% in control and 16% in transgenic) (Figure 4A,B). These results suggested that microglia cells were resting in control animals and activated in AD-mice.

After determining the occurrence of A_2A_-NMDA receptor complexes in microglia primary cultures, we moved to characterize the activated microglia phenotype in the APP_Sw,Ind_ mice model. When microglia cells are activated, they can evolve showing different characteristics, with two opposite phenotypes: M1 microglia inducing a pro-inflammatory state and cytotoxic effects and M2 microglia inducing an anti-inflammatory state and neuroprotective effects. Then, the microglia primary cultures of the control and APP_Sw,Ind_ mice models were prepared and analyzed by immunocytochemistry. Through the analysis, an important increase in inducible nitric oxide synthase (iNOS) a marker of the M1 phenotype, was observed in microglia from APP_Sw,Ind_ mice compared to controls (Figure 4C,D). Moreover, fluorescence signal quantitation indicated an important increase in arginase-1 (Arg-1), a marker of the M2 phenotype, in APP_Sw,Ind_ –derived microglia (Figure 4C,D). These results indicated an important increase in both M1 and M2 microglia in the AD-mice model. Interestingly, when the same assay was repeated by pretreating the APP_Sw,Ind_-derived cultures for one week with the A_2A_R antagonist SCH-58261, a significant decrease in iNOS and a small but significant increase in Arg-1 fluorescence were observed, indicating that A_2A_R antagonists skew activated microglia towards the neuroprotective M2 phenotype.

Then, no alteration in microglia proliferation marker was observed upon analysis of the Ki-67 fluorescence signal in the APP_Sw,Ind_ mice models treated or not with SCH-58261 and control (WT) animals (Figure 4C,D).

### 3.5. A_2A_R Activation Negatively Modulates NMDA Receptor Signaling in Microglia but it Increases NMDAR Signaling in Neurons from APP_Sw,Ind_ Mice

To analyze the adenosine regulatory effects over NMDAR functionality in the AD model, primary cultures of microglia and neurons were prepared from the hippocampus of APP_Sw,Ind_ mice and control animals. Intracellular cAMP levels, ERK1/2 phosphorylation and DMR were determined upon receptor activation and coactivation.

On the one hand, A_2A_R activation led to an increase in cAMP levels in the neurons and microglia from control and transgenic animals. As expected, NMDA did not modify cAMP levels but counteracted the action of A_2A_R agonists. In addition, the selective NMDAR antagonist reverted the effect of CGS-21680 (cross-antagonism) (Figure 5A–D). The MAPK pathway activation was similar in microglia from control and transgenic animals, with a similar effect in the individual activation or the coactivation of receptors. In these cells, bidirectional cross-antagonism was found (Figure 5E,F). In neurons coactivation led to a more robust effect than individual treatments. We found a partial cross-antagonism that fits with the hypothesis that, in neurons, not all NMDAR are directly interacting with the A_2A_R (Figure 5G,H). DMR recordings in glial cells were only obtained upon A_2A_R activation with NMDA being ineffective and with a lack or a small cross-antagonism. A relevant finding was underscored using neurons, because those from the control animals were much less responsive to CGS-21680 than those from the transgenic animals. Moreover, NMDA also induced a robust response in the cells from transgenic animals; however, coactivation did not lead to synergism or additive effects (Figure 5I–L).

## 4. Discussion

Adenosine A_1_ and A_2A_ receptors have deserved attention as potential targets to prevent neurodegeneration [29]. NMDA-induced preconditioning studies have led to controversial results on the usefulness of agonists or antagonists of the A_1_ receptors to afford neuroprotection (see [30] and references therein). In contrast, there is a consensus on the safety and the neuroprotective potential of the A_2A_ receptor antagonists. Pharmacological studies in parkinsonian models, the dopamine/adenosine antagonism and the use of knock-out mice reinforced the view that targeting the A_2A_R in striatal neurons could be useful in the therapy of Parkinson’s disease [31,32,33,34,35,36]. All the experimental effort plus drug discovery programs in pharmaceutical companies have led to the approval of istradefylline, a selective A_2A_R antagonist, for the therapy of Parkinson’s disease in Japan and the US [6,7,8].

Evidence on the possibility of targeting A_2A_R for combating dementia-related neurodegeneration has come from different sides and is consistent in both cell and animal models. Early synaptic deficits in the APP/PS1 mouse model of Alzheimer’s disease involved neuronal adenosine A_2A_R [37]. Electrophysiological studies in CA1 pyramidal neurons showed that the activation of those adenosine receptors enhanced chemically evoked NMDAR currents [38]. In an early β-amyloid-based AD model, adenosine production from ATP release was detrimental for cognition but not in the knock-out A_2A_R mouse [39]. Classically, the A_2A_R in neurons has been the focus in AD-related research. Another recent example is the involvement of the neuronal receptor in the memory deficits and synaptic loss in a tauopathy mouse model. Previously, it was demonstrated that receptor gene deletion was neuroprotective in the tauopathy model [12,40]. Remarkably, the A_2A_ receptor has a relevant function in activated microglia, which was found surrounding the pathological hallmarks of AD [41,42]. The activation of microglial A_2A_R enhances the production of proinflammatory mediators [43]. These results fit with the finding that the blockade of the microglial receptor reduces neuroinflammation and more importantly, may lead to an improvement of cognitive impairment [44]. Cunha and collaborators have compiled in different reviews the potential of targeting A_2A_R in both neurons and microglia to combat cognition and/or neurodegeneration in Alzheimer’s disease and other age-related dementias [45,46]. Coincidentally, it is well accepted that the most consumed adenosine receptor antagonists, caffeine in coffee and theophylline in tea, do protect against suffering from Alzheimer’s disease [47].

Our results show that further to the blockade of classical signaling mediated G_s_-coupled GPCRs, A_2A_R antagonists have the potential to combat AD by impacting one of the most relevant pathophysiological molecular mechanisms, namely modulation of NMDAR function. The early studies showing neuroprotection via adenosine impacting on NMDA-mediated effects [9] were never attributed to a direct interaction between adenosine and NMDA receptors. We here identified A_2A_-NMDA receptor complexes with particular properties. Unlike GPCRs that are prone to form complexes containing two different receptors, few examples of direct interactions between metabotropic (GPCR) and ionotropic receptors have been reported, probably due to lack of ad hoc assays. Interestingly, the cross-antagonism here detected for the A_2A_R-NMDAR couple was often found in GPCR-GPCR complexes; when cross antagonism is actually detected, GPCR-GPCR heteromer formation is suspected [25,26,48]. As further discussed below, the impairment of NMDAR function by A_2A_R antagonists is an attractive possibility to afford neuroprotection in AD.

The assessment of receptor complex expression led to various relevant findings. On the one hand, the expression in activated microglia was markedly higher than in resting microglia. Moreover, the expression in the microglia from the APP_Sw,Ind_ was higher than that in the microglia from the control mice. These results agree with a different phenotype of microglia in these transgenic AD models and reinforces the hypothesis that microglia from the APP_Sw,Ind_ protect neurons as the cognitive impairment only appears several months after birth [49]. On the other hand, complexes were also expressed in hippocampal neurons although the expression was similar in transgenic and control mice. By combining the expression data with the signaling results, it seems that the amount of A_2A_R-NMDAR interactions was comparatively higher in the microglia than in neurons and that there were a significant number of NMDAR in hippocampal neurons that were not interacting with the A_2A_R. The latter does not imply a lack of functional A_2A_R-NMDAR interactions but that both direct allosteric modulations within the macromolecular complex as well as indirect interactions, i.e., via cross-talk between second messengers, like Ca^2+^, impacting on the cAMP-PKA pathway.

As pointed out in the introduction, we undertook this investigation to find indirect ways to modulate the overactivity of the NMDAR. At least in part, neuronal death in AD (and in other neurodegenerative diseases) is due to excitotoxicity, i.e., by excess of glutamate that in turn results in the exacerbation of NMAR functionality [50]. Unfortunately, full blockade of the NMDAR is not feasible as it is fundamental for neural cell viability, while the current NMDAR allosteric modulators are not showing significant efficacy in either AD patients nor neuroprotection. Hence, current attempts to prevent neuronal death in AD should target either neurons, which are already altered, or glial cells surrounding “suffering” neurons. In our opinion, the results in this paper plus the literature data on neuroprotection by targeting the A_2A_R (see [29,51] for review) reinforces the hypothesis that A_2A_R antagonists could modulate glutamatergic action in AD and afford neuroprotection.

## Figures and Tables

**Figure 1 cells-09-01075-f001:**
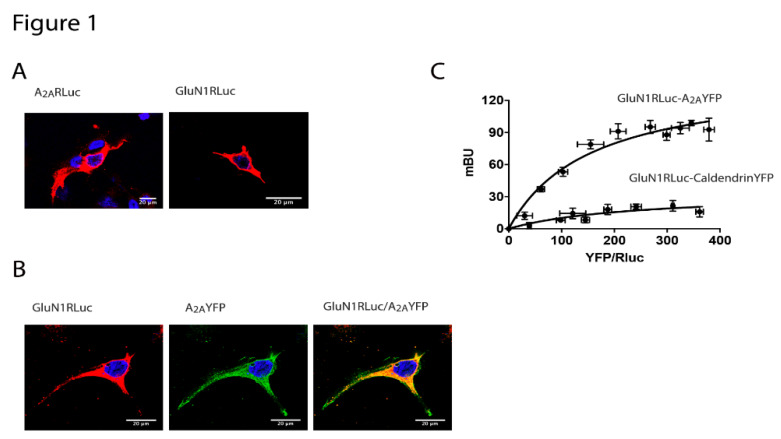
The *N*-methyl d-aspartate (NMDA) and the adenosine A_2A_ receptors interact to form heteromeric complexes. In (**A**,**B**), HEK-293T cells were transfected with 0.5 μg cDNA corresponding to the A_2A_ receptor (A_2A_R)-*Renilla* luciferase (RLuc) (**A left**) or 0.5 μg cDNA corresponding to the GluN1-RLuc in the presence of 0.3 μg cDNA corresponding to GluN2 (**A right**) or co-transfected with 0.4 μg cDNA for A_2A_R-yellow fluorescent protein (YFP) and 0.4 μg cDNA corresponding to the GluN1-RLuc in the presence of 0.25 μg cDNA corresponding to the GluN2 (**B**). Confocal microscopy images are shown. The receptors fused to RLuc were identified by immunocytochemistry (red) and the proteins fused to YFP were identified by its own fluorescence (green). Colocalization is shown in yellow in the merge image. Scale bar: 20 µm. In (**C**), the bioluminescence resonance energy transfer (BRET) saturation experiments were performed in HEK-293T cells transfected with 0.3 μg of cDNA corresponding to the GluN1-RLuc, 0.2 μg of cDNA corresponding to the GluN2 and increasing amounts of cDNA corresponding to A_2A_R-YFP (0.1 μg to 0.5 μg) or caldendrin-YFP (0.1 μg to 0.7 μg) as a negative control. The relative amount of BRET is given as a function of 1000× the ratio between the fluorescence of the acceptor (YFP) and the luciferase activity of the donor (RLuc). BRET is expressed as milli BRET units (mBU) and is given as the mean ± SEM of 6 different experiments grouped as a function of the amount of BRET acceptor.

**Figure 2 cells-09-01075-f002:**
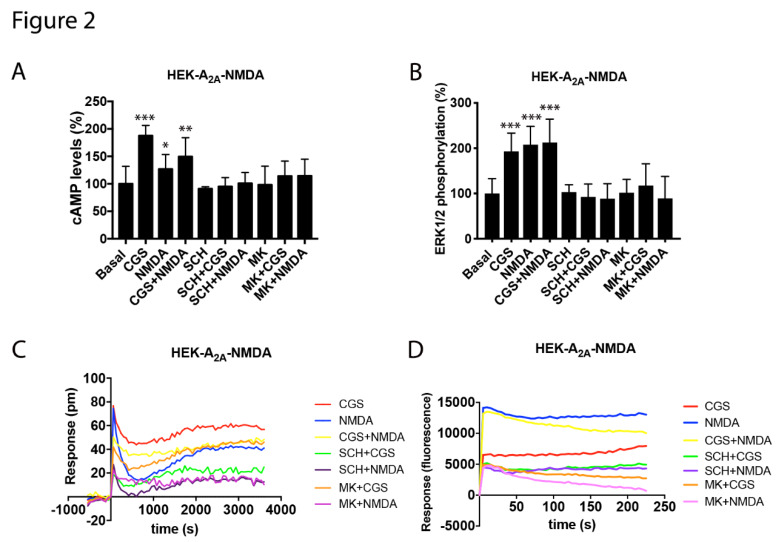
Functional characterization of the A_2A_-NMDA receptor heteromer in HEK-293T cells. In (**A**) to (**D**), HEK-293T cells expressing A_2A_R (0.5 μg of cDNA), GluN1 (0.5 μg of cDNA) and GluN2 (0.5 μg of cDNA) (**A**–**C**) or expressing A_2A_R (0.5 μg of cDNA), GluN1 (0.5 μg of cDNA), GluN2 (0.5 μg of cDNA) and 6GCaMP calcium sensor (0.75 μg of cDNA) (**D**) were pre-incubated or not with 1 μM of the A_2A_R antagonist SCH-58261 (SCH) or with 1 µM of the NMDA antagonist, MK-801 (MK), followed by treatment with 100 nM of the A_2A_R agonist CGS-21680 (CGS), 15 µM of NMDA or both, and the cAMP levels (**A**), extracellular signal-regulated (ERK) 1/2 phosphorylation (**B**), representative traces of dynamic mass redistribution (DMR) (**C**) and the representative traces of intracellular Ca^2+^ responses over time (**D**) were determined. Values are the mean ± SEM of 10 to 12 different experiments. ERK 1/2 phosphorylation levels and cAMP increases are expressed as percentage over basal. A one-way ANOVA followed by a Bonferroni multiple comparison post-hoc test showed a significant effect over 100% (* *p* < 0.05, ** *p* < 0.01, *** *p* < 0.001).

**Figure 3 cells-09-01075-f003:**
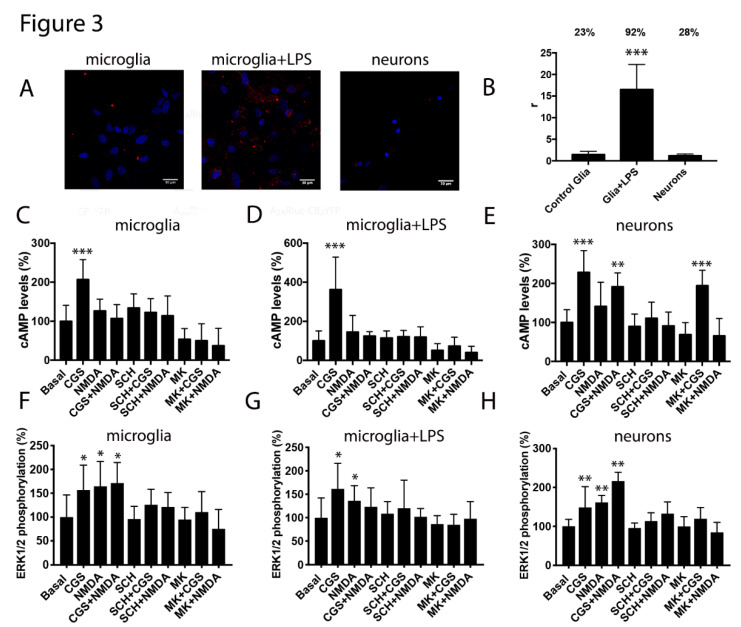
Expression and functionality of the A_2A_ and NMDA receptors heteromers in the microglial and neuronal primary cultures. Panels (**A**,**B**). The proximity ligation assay (PLA) was performed in mice microglial primary cultures treated or not with 1 µM lipopolysaccharide (LPS) and 200 U/mL interferon-γ (IFN-γ) to activate the microglia and in the mice hippocampal primary cultures of neurons using the primary antibodies specific for the A_2A_ and NMDA receptors. In all cases, the cell nuclei were stained with Hoechst (blue). The confocal microscopy images are shown (superimposed sections) in which heteromers appear as red clusters (in neurons or microglia). Scale bars = 30 μm. The bar graph (**B**) shows the number of the red dots/cell (r) and the numbers above the bars indicate the percentage of cells presenting red dots. Values are the mean ± SEM (*n* = 8). A one-way ANOVA followed by Bonferroni’s multiple comparison post-hoc test were used for statistical analysis (*** *p* < 0.001, versus control -resting- microglia). Panels (**C**–**H**) display the microglial primary cultures treated (**D**,**G**) or not (**C**,**F**) with 1 µM LPS and 200 U/mL IFN-γ and the mice hippocampal primary cultures of the neurons (**E**,**H**) that were pre-incubated or not with 1 μM of the A_2A_R antagonist SCH-58261 (SCH) or with 1 µM of the NMDAR antagonist MK-801 (MK) followed by treatment with 100 nM of the A_2A_R agonist CGS-21680 (CGS), 15 µM NMDA or both, and the cAMP levels (**C**–**E**) and ERK 1/2 phosphorylation signal (**F**–**H**) were determined. The values are the mean ± SEM of 10 to 12 different experiments. The ERK 1/2 phosphorylation levels and cAMP increases were expressed as a percentage over basal. A one-way ANOVA followed by a Bonferroni multiple comparison post-hoc test showed a significant effect over 100% (* *p* < 0.05, ** *p* < 0.01, *** *p* < 0.001).

**Figure 4 cells-09-01075-f004:**
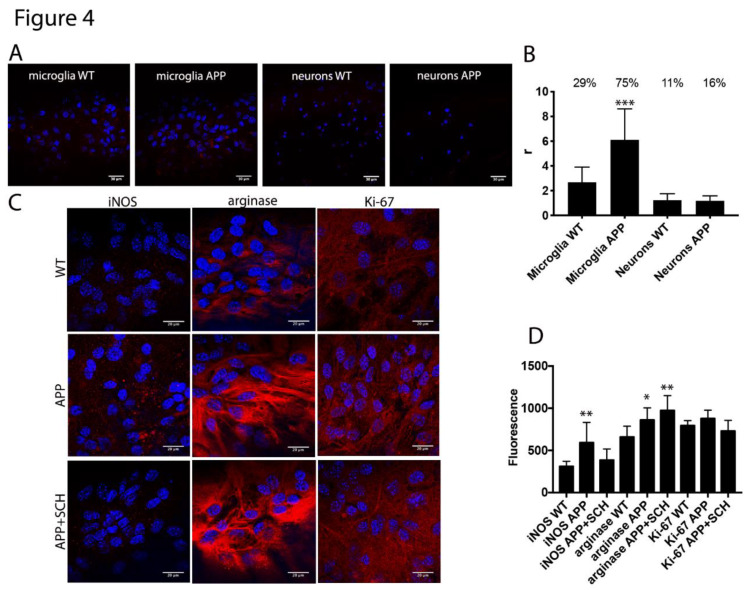
Microglial markers in primary cultures from APP_Sw,Ind_ mice. In (**A,B**), proximity ligation assays (PLAs) were performed in microglial and in the hippocampus neuronal primary cultures of control (WT) and APP_Sw,Ind_ mice using primary antibodies specific for A_2A_ and NMDA receptors. In all cases, the cell nuclei were stained with Hoechst (blue). The confocal microscopy images are shown (superimposed sections) in which the heteromers appear as red clusters (in neurons or microglia). Scale bars = 30 μm (neurons and microglia). The bar graph (**B**) shows the number of red dots/cell (r) and the numbers above the bars indicate the percentage of cells presenting red dots. Values are the mean ± SEM (*n* = 6). A one-way ANOVA followed by Bonferroni’s multiple comparison post-hoc test were used for statistical analysis (*** *p* < 0.001, versus WT -control-). In (**C**,**D**), immunocytochemical assays were performed in the primary cultures of microglia APP_Sw,Ind_ mice or control (WT) animals pretreated or not for one week with the A_2A_R antagonist SCH-58261. The staining was performed using the antibodies that detected either arginase-1 marker, the antiproliferation cell protein Ki-67 or nitric oxide synthase and a Cy3-conjugated secondary antibody (red). The fluorescence was quantified in all the panels using the Fiji program and normalization by cell number. Representative images in all conditions are shown in (**C**). A one-way ANOVA followed by Bonferroni’s multiple comparison post-hoc test were used for statistical analysis (* *p* < 0.05, ** *p* < 0.01; versus WT).

**Figure 5 cells-09-01075-f005:**
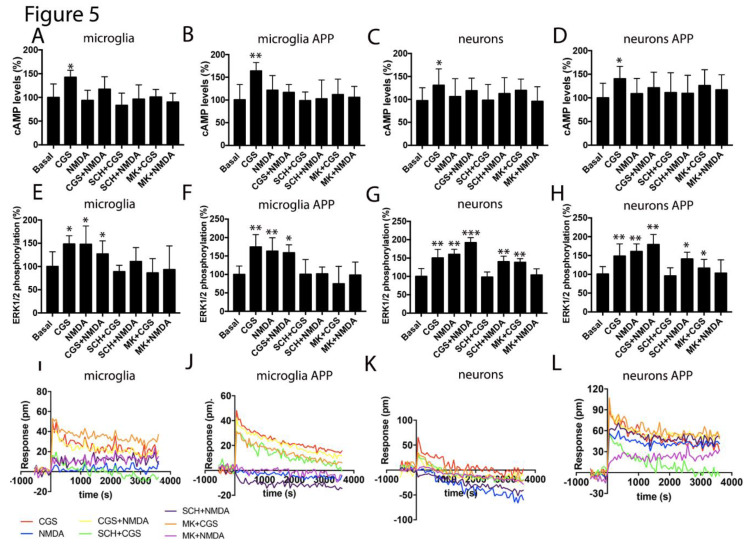
Functionality of the A_2A_-NMDA receptor heteromer in the APP_Sw,Ind_ mice model of Alzheimer’s disease(AD). In (**A**–**L**), the microglial and neuronal primary cultures from the hippocampus of the WT and APP_Sw,Ind_ mice (**E**,**H**) were pre-incubated with vehicle, 1 μM of the A_2A_R antagonist SCH-58261 (SCH) or 1 µM of the NMDAR antagonist, MK-801 (MK), followed by treatment with 100 nM of the A_2A_R agonist CGS-21680 (CGS), 15 µM NMDA or both, and the cAMP levels (**A**–**D**), ERK 1/2 phosphorylation signal (**E**–**H**) and the representative traces of dynamic mass redistribution (DMR) over time (**I**–**L**) were determined. Values are the mean ± SEM of 10 to 12 different experiments. ERK 1/2 phosphorylation levels and cAMP increases are expressed as percentage over basal. A one-way ANOVA followed by a Bonferroni multiple comparison post-hoc test showed a significant effect over basal (* *p* < 0.05, ** *p* < 0.01, *** *p* < 0.001).

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
