# Peer review of "Adenosine A2A Receptor Antagonists Affects NMDA Glutamate Receptor Function. Potential to Address Neurodegeneration in Alzheimer’s Disease"

_cells, 2020, doi:10.3390/cells9051075_

Round 1

Reviewer 1 Report

This is very interesting work. Authors described mechanism of metabotropic A2A and ionotropic NMDA receptor interaction and propose that this interaction may be used in treatment of Alzheimer disease. Although the study is performed in vitro (neuronal and microglial primary cultures) the results are very convincing and presented in elegant way. Perfect work!

Author Response

We appreciate these comments.

Reviewer 2 Report

The paper entitled “ Adenosine A2A receptor antagonists affects NMDA glutamate receptor function. Potential to address neurodegeneration in Alzheimer’s disease” by Franco et al. regards the study of influence of adenosine A2A receptor in NMDAR functionality. It is well written and arranged. In my opinion it is suitable for publication in Cells journal after minor revisions.

General comments

In the abstract (line 28) the  authors when write about receptors should explain well which receptor intend  A2A or NMDA.  In addition they should indicate which  kind of antagonists were approved for Parkinson’s disease. 

In the introduction (line 51) the authors should explain which type of allosteric modulators of the NMDA were used. I mean if these allosteric modulators were positive or negative and they should insert the reference. In the tex Gs should be change in Gs

 In Materials and Methods they should insert   “all animal procedures were performed in accordance with the European Community guidelines…”

In the captions under the figures, it is reported a wrong p value (see figures 2, 3, 5, and figure A1 in supplementary section). I mean “**p < 0.01. ***p < 0.01” should be change in “ **p < 0.01. ***p < 0.001”

Minor revisions:

page 2, line 69 “studies made in to explore…” should be changed in “studies conducted to explore…”;

page 3, line 83 “….is also expressed [13-16] Our results…” should be changed in “….is also expressed [13-16]. Our results…”;

page 4, lines 120-123 these two sentences should be rearranged without repetition;

page 8, line 293 “Fig. 1A” should be changed in “Figure A1 in supplementary section”.

Author Response

We appreciate the comments.

All issues have been addressed in the revised version of the manuscript